# VIDEO ANOMALY DETECTION VIA SINGLE FRAME SUPERVISION

## ABSTRACT

Video Anomaly Detection (VAD) aims to identify anomalous frames in given videos. Existing fully-supervised VAD encounters substantial annotation cost and weakly-supervised VAD suffers from the deficiency of weak labels. In this paper, we propose a more effective Single Frame supervised VAD (SF-VAD), which leverages single abnormal frame as label. We argue that single abnormal frame annotation is highly efficient and it provides fine-grained dual references to abnormal and normal frames, which facilitate dependable anomaly and normality decoupling. Leveraging these dual references, we propose Frame-guided Probabilistic Contrary Learning (FPCL), to decouple contrary abnormal and normal patterns. Specifically, FPCL learns inclusive abnormal patterns reliably from mined abnormal frames, guided by similarity-based abnormal probability. Simultaneously, it decouples normal patterns in abnormal videos, by learning normal patterns in preceding frames, guided by Gaussian-prior normal probability. In inference, we additionally design temporal decoupling and boundary refining modules to reveal discriminative abnormal characters of temporal features. Extensive experiments show our SF-VAD method outperforms SOTA methods and achieves an optimal performance-cost trade-off. To support future research on this topic, we construct and release three SF-VAD datasets.

## 1 INTRODUCTION

Video Anomaly Detection (VAD) strives to identify anomalous frames in given videos, which draws substantial research interest for its practical applications (Sultani et al., 2018; Tian et al., 2022). By supervision paradigm, existing VAD research can be categorized into three classes, as in Fig. 1a. Intuitively, pioneer works (Landi et al., 2019; Liu & Ma, 2019) explore Fully-supervised VAD (F-VAD), utilizing labels of each frame. However, due to the ambiguity of anomaly boundary, F-VAD encounters the issue of human bias and its annotation is also labor-intensive. Alternatively, researchers (Gong et al., 2019; Yang et al., 2022) study Semi-supervised VAD (S-VAD) where only normal patterns are learned during training, and, out-of-distribution frames are identified as abnormal. Unfortunately, it is prone to recognize unseen normal patterns as anomalous, encountering severe false alarms. To address these drawbacks, recent studies (Wu et al., 2023; Zhang et al., 2023; Chen et al., 2024) focus on Weakly-supervised VAD (W-VAD), which delivers a superior performance-cost trade-off by leveraging video-level weak labels.

Current W-VAD methods (Sultani et al., 2018; Yu et al., 2022; Cho et al., 2023; Chen et al., 2024; Feng et al., 2021a; Li et al., 2022) learn abnormal patterns in abnormal videos by frames that have higher predicted anomaly scores, and learn normal patterns by such frames in normal videos. However, W-VAD confronts fundamental dilemmas due to the deficiencies of video-level weak labels. First, when learning abnormal patterns, weak labels lack fine-grained references to abnormal frames. This leads the learning of anomaly to rely on groundless and noisy frame-level predictions, resulting in unreliable anomaly modeling. Second, when learning normal patterns, weak labels lack reference to normal frames in abnormal videos. Consequently, the learning of normality omits these normal frames, which are then misinterpreted as anomalous, causing false alarms.

To overcome these deficiencies, we take an intuitive step towards more effective supervision for VAD. In real-life scenarios, when humans identify an abnormal video, it is unnecessary to analyze the entire video — instead, all they need to do is identify an abnormal frame within the video. We

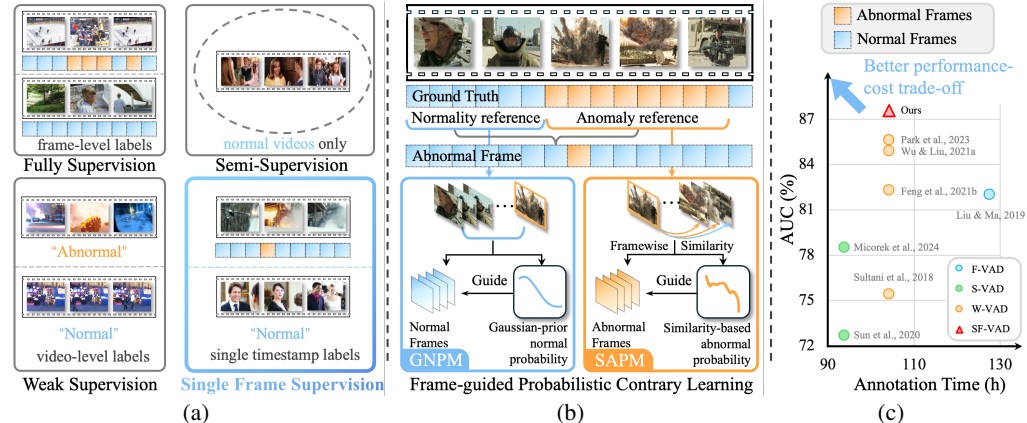

Figure 1: (a) Overview of existing supervisions of VAD and Single Frame Supervision. (b) Illustration of anomaly and normality references of SF-VAD, and concept of frame-guided probabilistic contrary learning. (c) AUC Performance comparison (%) versus annotation time (hours) on UCF-Crime dataset. SF-VAD shows an optimal performance-cost trade-off over other methods (details on annotation time estimation are in Appendix E).

argue this abnormal frame offers more informative supervision, and its annotation is highly efficient. Thus, we propose *Single Frame supervised VAD* (SF-VAD), as in Fig. 1a. Correspondingly, we construct three SF-VAD datasets, SF-ShanghaiTech, SF-UCF-Crime, SF-XD-Violence, by re-annotating widely adopted VAD datasets (Liu et al., 2018a; Sultani et al., 2018; Wu et al., 2020).

SF-VAD is a VAD paradigm that leverages each abnormal video with an abnormal frame label, and each normal video with a video-level normal label. Besides access to a specific abnormal frame, SF-VAD further provides insightful dual references to decouple contrary patterns, as in Fig. 1b. On the one hand, SF-VAD offers precise anomaly reference, which can be utilized to mine congruent abnormal frames, facilitating accurate and comprehensive anomaly modeling. On the other hand, SF-VAD implies normality reference in abnormal videos. As annotators typically mark close to the beginning of abnormal events for efficiency, preceding frames can reasonably be considered as normal. This provides a credible reference to decouple normal patterns in abnormal videos.

Leveraging dual references, we propose Frame-guided Probabilistic Contrary Learning (FPCL), to model contrary anomalous and normal patterns simultaneously, thereby decoupling these opposite patterns. Concretely, FPCL models opposite patterns by two modules. First, we devise a Similarity-based Abnormal Pattern Modeling (SAPM) module, to model inclusive abnormal patterns reliably. SAPM first mines the similar abnormal frames by exploring frame-wise similarity between labeled abnormal frame and frames in consecutive interval. As similar abnormal frames may contain noise, SAPM integrates similarity-based abnormal probability to guide the anomaly learning from mined frames, enabling a dependable learning process. Likewise, we introduce Gaussian-prior Normal Pattern Modeling (GNPM) module, to decouple the normal patterns in abnormal videos. Treating preceding frames away from abnormal frame as normal, it learns credible normal patterns, guided by Gaussian-prior normal probability. During inference, we additionally devise the temporal decoupling and boundary refining modules, which calibrate attention map to disentangle abnormal and normal temporal dependencies and reveal discriminative temporal character. Extensive experiments show our method outperforms state-of-the-art VAD methods and achieves an optimal performance-cost trade-off, as depicted in Fig. 1c. Our main contributions are summarized as follows:

- We propose an effective Single Frame supervised VAD paradigm, which provides insightful dual references to abnormal and normal frames at negligible extra cost. To support future research on this topic, we construct and release three SF-VAD datasets.

- We devise frame-guided probabilistic contrary learning, to decouple contrary anomalous and normal patterns via single frame supervision. In inference, we tailor temporal decoupling and boundary refining modules to reveal discriminative temporal character.

- Extensive results show our SF-VAD method surpasses previous state-of-the-art VAD methods, including a higher AUC performance and a lower false alarm rate.

## 2 BACKGROUND

### 2.1 WEAKLY-SUPERVISED VAD

In Weakly-supervised VAD (W-VAD), only binary video-level labels are available during training, as illustrated in Fig. 1a. To identify anomalous frames from given video $\mathcal{X}$, W-VAD methods predict frame-level anomaly scores $\tilde{\mathcal{Y}}$. Sultani et al. (2018) initially introduce Multi-Instance Learning (MIL) framework to W-VAD, owing to its ability to discriminate abnormal and normal patterns by weak labels. Specifically, MIL encourages the top anomaly scores in an abnormal video to be higher than those in a normal video by MIL-based loss:

$$\mathcal{L}_{\text{MIL}} = -\log(\frac{1}{k} \sum_{i \in \text{top-k}} \tilde{\mathcal{Y}}_i^a) - \log(1 - \max(\tilde{\mathcal{Y}}^n)) \tag{1}$$

where $i$ is index of frames and $a$, $n$ indicate abnormal and normal videos respectively. Following MIL framework, some works refine the learning framework. Lv et al. (2023) address the context bias in MIL by clustering the ambiguous frames and reforming MIL to leverage the clustered frames for bias mitigation. Cho et al. (2023) propose relative distance learning to enlarge the gap of anomaly and normality features. Some researchers employ two-stage training strategy to alleviate noise. Feng et al. (2021b); Zhang et al. (2023) utilize two-stage self-training strategy where the pseudo labels are generated in the first stage by MIL and the discriminative features are learned by pseudo labels in the second stage. Some works use prompt learning to learn abnormal semantics. Chen et al. (2024) introduce text prompts and normal context prompts to learn discriminative anomaly representations. Wu et al. (2023) leverage expressive text features to learn the semantics of various abnormal scenes.

### 2.2 INEXACT SUPERVISION

Single frame supervision falls under the category of inexact supervision, which relates to learning from imprecisely labeled data. Inexact supervision is widely applied to diverse computer vision tasks due to its promising performance and minor annotation requirements. In semantic segmentation, Bearman et al. (2016) first introduce inexact supervision, by applying effective point supervision, which delivers a promising performance. In video spatio-temporal action localization, Mettes et al. (2016) extend inexact supervision to frame supervision. Later, Ma et al. (2020) introduce frame supervision to temporal action localization and a corresponding SF-Net that achieves a competitive performance against fully-supervised methods. Li et al. (2021) propose a temporal action segmentation method that integrates model prediction with annotated timestamps to boost accurate segmentation. Recently, Cui et al. (2022) introduce frame supervision to language-driven moment retrieval, which uses the Gaussian distribution to model the probability distribution of foreground action behaviors. In VAD task, inexact supervision has not been yet explored. Therefore, we propose single frame supervised VAD to achieve an optimal performance cost trade-off.

## 3 METHOD

### 3.1 SINGLE FRAME SUPERVISED VAD

Single Frame supervised VAD (SF-VAD) utilizes the abnormal frame label $t$ to learn abnormal patterns. Label $t$ refers to the index of a frame in the given video $\mathcal{V}$ where abnormal behavior is observed. Under SF-VAD, a training sample is $(\mathcal{V}^a, t)$ for each abnormal video, or $(\mathcal{V}^n, 0)$ for each normal video, where 0 indicates its normality. To identify the anomalous frames, SF-VAD models predict frame-level anomaly scores, where 0 denotes normal and 1 refers to abnormal.

### 3.2 BASELINE

The architecture of the framework is depicted in Fig. 2. First, the input media is split into 16-frame non-overlapping clips owing to the redundancy of adjacent frames. Pre-trained encoders are utilized to extract embeddings. Then, multi-modal embeddings are then concatenated along the dimension as $\mathcal{X} \in \mathbb{R}^{L \times D_m}$ where $L$ equals the number of the clips and $D_m$ is the dimension of the features.

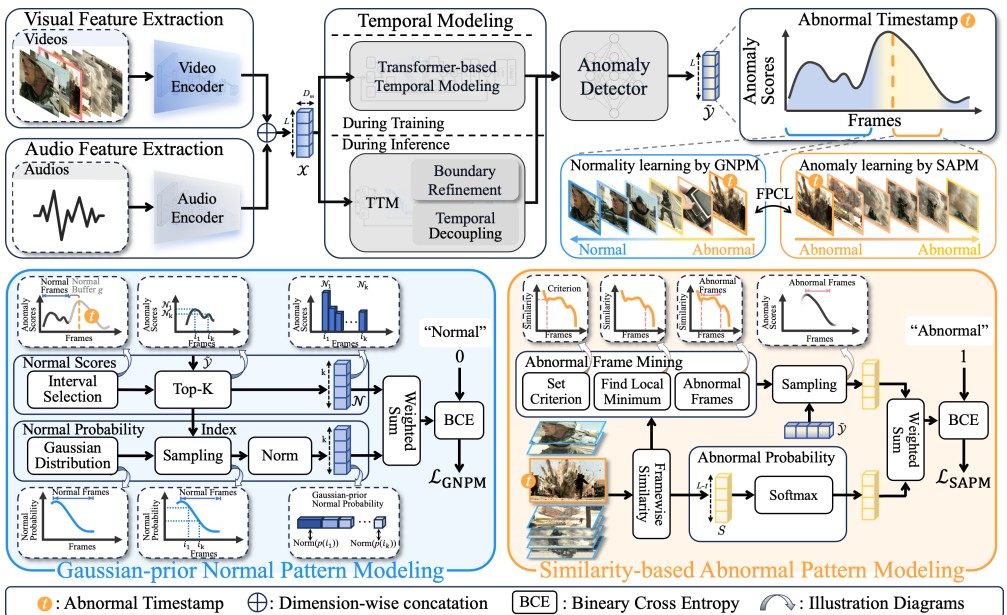

Figure 2: Overview of proposed method. Multi-modal features $\mathcal{X}$ are extracted by encoders. Then, it passes through temporal modeling and anomaly detector to generate frame-level anomaly scores $\widetilde{\mathcal{Y}}$. Abnormal patterns are modeled by SAPM, which learns from mined abnormal frames, guided by similarity-based abnormal probability. Correspondingly, normal patterns are decoupled by GNPM, which models from preceding normal frames, guided by Gaussian-prior normal probability.

**Transformer-based Temporal Modeling** (TTM) module utilizes Transformer (Vaswani et al., 2017) architecture, which has been verified as a highly effective architecture for modeling temporal relationships. Thus, we employ TTM to capture multi-scale temporal dependencies, following (Pu et al., 2024). In essence, TTM first computes attention map $\mathcal{M} \in \mathbb{R}^{L \times L}$ after the linear projection of input feature $\mathcal{X}$. Subsequently, to magnify short-term dependencies, $\mathcal{M}$ is masked into a local attention map $\widetilde{\mathcal{M}} \in \mathbb{R}^{L \times L}$. Then, global features $\mathcal{F}$ and local features $\widetilde{\mathcal{F}}$ are calculated based on the corresponding attention map. Subsequently, long-term and short-term dependencies are integrated by a gate parameter $\alpha$. Finally, temporal features $\mathcal{F}^c \in \mathbb{R}^{L \times D_m}$ are obtained through a Multi-Layer Perceptron (MLP). A detailed description of the TTM can be found in Appendix A.

**Anomaly Detector** is a MLP-based network to acquire frame-level anomaly scores $\widetilde{\mathcal{Y}} \in \mathbb{R}^L$ by temporal features. It can be denoted as:

$$
\begin{aligned}
\text{MLP} &= \text{Dropout}\left(\text{GELU}\left(\text{Conv}\left(\cdot\right)\right)\right) \\
\widetilde{\mathcal{Y}} &= \sigma\left(f_t(\text{MLP}(\text{MLP}(\mathcal{F}^c)))\right)
\end{aligned}
\tag{2}
$$

where $\text{Conv}(\cdot)$ refers to one-dimension convolution followed by GELU (Hendrycks & Gimpel, 2016) and $f_t(\cdot)$ represents causal convolutional layer. $\sigma$ indicates the sigmoid activation function.

### 3.3 FRAME-GUIDED PROBABILISTIC CONTRARY LEARNING

Frame-guided Probabilistic Contrary Learning (FPCL) models abnormal patterns, while decoupling normal patterns in abnormal videos simultaneously by proposed similarity-based anomaly pattern modeling and Gaussian-prior normal pattern modeling modules.

**Multi-Instance Learning** is employed by most works to distinguish abnormal frames, as in Sec. 2.1. It can be fitted to SF-VAD by replacing top-scoring operation with the abnormal frame label $t$:

$$
\mathcal{L}_{\text{MIL}} = -\log(\mathcal{Y}_t^a) - \log(1 - \max(\widetilde{\mathcal{Y}}^n))
\tag{3}
$$

However, such an intuitive approach is insufficient for capturing the full spectrum of abnormal behaviors, and it neglects the normal patterns in abnormal videos.

---

**Algorithm 1** Abnormal Frame Mining

---

**Input:** Abnormal frame $t$, Similarity matrix $S$, Threshold $\hat{\theta}$
**Output:** Set of abnormal frames $\mathcal{T}$
 1: Initialization: $i \leftarrow t, \mathcal{T} \leftarrow \{\}$
 2: $i \leftarrow \min_{j>t}(j \mid S_j < S_{j-1} \text{ and } S_j < S_{j+1})$ // Find local minimum index
 3: **if** $1 - S_i < \hat{\theta}$ **then**
 4:     $p \leftarrow 1 - S_i$ // Set criterion $p$
 5:     $\mathcal{T} \leftarrow \mathcal{T} \cup \{j | j \geq t \text{ and } j \leq i\}$
 6:     **repeat**
 7:         $i \leftarrow \min_{j>i}(j \mid S_j < S_{j-1} \text{ and } S_j < S_{j+1})$
 8:         $l \leftarrow \max\{S_j \mid S_j > S_i, j \in [t+1, i-1]\}$ // Find left base index $l$ of $S_i$
 9:         $r \leftarrow \max\{S_j \mid S_j > S_i, j \in [i+1, L]\}$ // Find right base index $r$ of $S_i$
10:         $\hat{p} \leftarrow S_i - \min(S_l, S_r)$ // Calculate prominence
11:         **if** $\hat{p} > p$ **then**
12:             $\mathcal{T} \leftarrow \mathcal{T} \cup \{i\}$
13:         **end if**
14:     **until** $\hat{p} > p$
15: **else**
16:     $\mathcal{T} \leftarrow \{t\}$ // Exit mining
17: **end if**

---

**Similarity-based Abnormal Pattern Modeling** (SAPM) module is proposed to learn inclusive abnormal patterns reliably by single frame supervision. Specifically, SAPM consists of two steps which are abnormal frame mining, and subsequent abnormal pattern modeling, guided by similarity-based abnormal probability.

Abnormal frame $t$ indicates the specific abnormal behavior. However, such abnormal behavior is not intact, limited by the volume of a single frame. Besides, we discover that the learning process is extremely sensitive to noise when modeling abnormal patterns. Therefore, we strive to ensure an accurate anomaly pattern learning from inclusive and reliable frames, which benefits the detection of multiple abnormal events with similar patterns. To this end, regarding abnormal frame $t$ as the anomaly reference, we design an abnormal frame mining algorithm to mine similar abnormal frames based on the consistency of frame-wise cosine similarity, as depicted in Algorithm 1. First, the cosine similarity is calculated between the feature of $t$ and adjacent features after $t$:

$$S = \left[ \frac{\mathcal{X}_t \cdot \mathcal{X}_i}{\|\mathcal{X}_t\|_2 \|\mathcal{X}_i\|_2} \;\middle|\; i \in [t, L] \right] \tag{4}$$

where $S \in \mathbb{R}^{L-t}$ indicates the feature similarity and $\| \cdot \|_2$ refers to 2-Norm. Our assumption is following frames with a higher coherency of similarity show a greater likelihood to be within the abnormal interval of $t$. The first local minimum of the similarity is set as a criterion. The abnormal frames are in the interval where frame-wise similarity is steady. The set of abnormal frame index is denoted as $\mathcal{T}$. If the similarity decreases greatly beyond threshold $\hat{\theta}$, the coherence of the interval is sub-optimal for reliable abnormal frame mining. Thereby, the mining process exits.

Afterward, the intact abnormal patterns are learned dependably with similarity-based abnormal probability. The computation of $\mathcal{L}_{\text{SAPM}}$ can be denote as:

$$\mathcal{L}_{\text{SAPM}} = -\log\left(\sum_{i \in \mathcal{T}} \widetilde{\mathcal{Y}}_i \frac{\exp(S_i)}{\sum_{j \in \mathcal{T}} \exp(S_j)}\right) \tag{5}$$

**Gaussian-prior Normal Pattern Modeling** (GNPM) module is devised to decouple normality in abnormal videos, thereby exaggerating abnormal frames. Concretely, GNPM learns normal patterns in preceding frames of $t$, guided by Gaussian-prior normal probability.

We regard frames before $t$ as normal frames. As mentioned above, the learning process is sensitive to noisy patterns, hence we set a normal buffer $g$ before abnormal frame $t$, considering annotation variance. The overall indices of the normal frame are from 1 to $t - g$. To focus on prominent normal features instead of background noisy features, we select top-k frames from the normal interval, and

k is set according to the normal interval length. The process can be denoted as:

$$\mathcal{N} = \left\{ \widetilde{\mathcal{Y}}_i \mid i \in \text{argtop-k}(\{\widetilde{\mathcal{Y}}_j \mid j \in [1, t-g]\}) \right\}, \text{ where k} = \left\lfloor \frac{t-g}{\eta} \right\rfloor \tag{6}$$

where $\mathcal{N}$ is the set of sampled normal scores and $\eta$ refers to a hyperparameter that controls $k$. To decouple normal behaviors for fine-grained anomaly detection, GNPM incorporates normal frames with normal probability. The prior is frames that are distant to abnormal frame $t$ are more likely to be normal frames, which is coherent to dataset statistics, as in Sec. 4.2. Mathematically, the character of Gaussian distribution is consistent with normal probability. Therefore, the normal probability is calculated by the distance to $t$ with Gaussian distribution. Normal probability is calculated by:

$$p(i) = \frac{1}{\sqrt{2\pi}\sigma} \exp\left(-\frac{i^2}{2\sigma^2}\right) \tag{7}$$

where $i$ is the index in normal collection and $\sigma$ is the variance of Gaussian distribution. In conclusion, $\mathcal{L}_{\text{GNPM}}$ can be computed as:

$$\mathcal{L}_{\text{GNPM}} = -\log(1 - \sum_{i=1}^{k} \mathcal{N}_i \frac{p(i)}{\sum_{j=1}^{k} p(j)}) \tag{8}$$

Consequently, the overall loss function for abnormal videos is $\mathcal{L} = \mathcal{L}_{\text{SAPM}} + \mathcal{L}_{\text{GNPM}}$.

### 3.4 INFERENCE

We observe that the global attention mechanism in the TTM module delivers noisy temporal features. In this case, we calibrate the attention map during inference to reveal discriminative temporal character with the proposed temporal decoupling module and boundary refining module.

**Temporal decoupling module** is designed to mitigate the obscure noise caused by the attention mechanism. The idea is the queries tend to gain more attention scores for abnormal frames, which leads to high variance in attention map. Thus queries with low variance in attention map are not located in abnormal intervals. The query-wise variance of the local attention map is computed as:

$$l = \max\left(0, i - \left\lfloor \frac{w}{2} \right\rfloor\right), r = \min(i + \lfloor \frac{w}{2} \rfloor, L)$$
$$\mu_i = \frac{1}{r-l} \sum_{j=l}^{r} \widetilde{\mathcal{M}}_{i,j} \tag{9}$$

where $\mu \in \mathbb{R}^L$ indicates the query-wise mean in the local attention map and $l, r \in \mathbb{R}$ refers to the left and right boundary of the local attention map $\widetilde{\mathcal{M}}$. Other notations can refer to Appendix A. The variance can be computed as:

$$\sigma_i^a = \frac{1}{r-l} \sum_{i=l}^{r} (\widetilde{\mathcal{M}}_{i,j} - \mu_i) \tag{10}$$

where $\sigma^a \in \mathbb{R}^L$ refers to the query-wise variance within the valid range of local attention map $\widetilde{\mathcal{M}}$. Then the attention of the query with a low variance is masked as follows:

$$\widetilde{\mathcal{M}}_{i \in \{\sigma_i < \theta\}, *} = 0 \tag{11}$$

where $\theta$ corresponds to the threshold value.

**Boundary refining module** is proposed which masks the obscure temporal features that lie outside the abnormal interval to generate clear event boundaries. First, we compute the key-wise sum in local attention map $\widetilde{\mathcal{M}}$ as denoted:

$$\hat{\mu}_j = \sum_{j=l}^{r} \widetilde{\mathcal{M}}_{i,j} \tag{12}$$

where $\hat{\mu} \in \mathbb{R}^L$ refers to the key-wise sum in local attention map. Then the local maximum values before and after the abnormal interval videos are selected as follows:

$$\epsilon = \text{argmax}_{i \in [1, \lfloor L/\theta' \rfloor]}(\hat{\mu}), \quad \hat{\epsilon} = \text{argmax}_{i \in [\lfloor L/\theta'' \rfloor, L]}(\hat{\mu}) \tag{13}$$

where $\epsilon$ and $\hat{\epsilon}$ indicate possible start and end points of the abnormal interval. $\theta'$ and $\theta''$ refer to interval threshold. Then the corresponding temporal feature $\mathcal{F}^t$ is masked accordingly, which can be denoted as:

$$\mathcal{F}^t_{i\in[1,\epsilon],j\in[\hat{\epsilon},L]} = 0 \tag{14}$$

## 4 EXPERIMENTAL RESULTS

### 4.1 DATASET CONSTRUCTION

To validate the feasibility of SF-VAD, we construct three high-quality human annotated SF-VAD datasets based on public VAD datasets (Sultani et al., 2018; Wu et al., 2020; Liu et al., 2018a). Annotations are obtained via a crowd-sourcing platform, where annotators, after passing a qualification test, are tasked with: 1) watching the video to identify an abnormal frame; 2) annotating the first abnormal frame they witness. Details of the dataset construction are depicted in Appendix C.

### 4.2 DATASET STATISTIC

In Fig. 3a, we present the Kernel Density Estimation (KDE) plot of the relative positions of annotated frames in abnormal videos, calculated as the ratio of the annotated frame index to the total number of frames. The distributions show a clear Gaussian-like pattern, peaking in the first half of the video. This validates the efficiency of single frame annotations, as annotators tend to focus on the earlier part of the video, reducing the need to review the entire footage. It also suggests that the preceding frames of the video, having a lower likelihood of being annotated, tend to be normal.

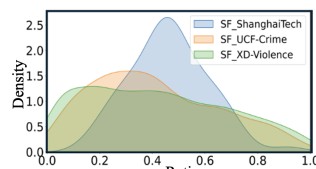

(a) KDE of relative positions of annotated frames in videos

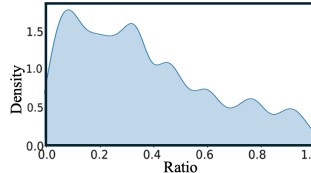

(b) KDE of relative positions of annotated frames in anomalies

Figure 3: Visualization of dataset statistic

Furthermore, we analyze the temporal characters of SF-VAD, utilizing frame-level labels of UCF-Crime, re-annotated by Liu & Ma (2019). In Fig. 3b, we visualize the relative position of these annotations within the abnormal intervals. The KDE curve manifests that anomalies are more frequently labeled toward the beginning of anomalies. This observation aligns with the anomaly reference assumption that similar abnormal frames tend to be contained in the following contiguous frames. In addition, it hints normality reference that preceding normal frames are typically in close proximity to the annotated abnormal frame. These temporal characters provide insightful references to decouple contrary patterns.

### 4.3 DATASET DESCRIPTION

XD-Violence (Wu et al., 2020) is the largest multi-modal VAD dataset containing 4754 untrimmed videos with a total duration of 217 hours. ShanghaiTech Campus (Liu et al., 2018a) comprises 437 videos from 13 fixed-view surveillance cameras. UCF-Crime (Sultani et al., 2018) comprises 1900 videos collected from various sources including videos from surveillance cameras and social media with a total duration of 128 hours. Detailed dataset description can be found in Appendix D.

### 4.4 EVALUATION METRICS

Following evaluation metrics of existing works (Sultani et al., 2018; Zhang et al., 2023), we employ the Area Under the Curve (AUC) for SF-ShanghaiTech and SF-UCF-Crime, and Average Precision (AP) for SF-XD-Violence. Furthermore, False Alarm Rate (FAR) with a threshold value of 0.5 is evaluated. A lower FAR demonstrates that the model can enable more reliable anomaly detection.

### 4.5 IMPLEMENTATION DETAILS

Consistent with current methods (Sultani et al., 2018; Cho et al., 2023), we utilize I3D (Carreira & Zisserman, 2017) video encoder to extract video features and VGGish (Hershey et al., 2017) audio

Table 1: Performance comparison with SOTA methods on three datasets.

| Supervision | Methods | Text Annotation | Feature | XD(%) | SH(%) | UCF(%) |
|---|---|---|---|---|---|---|
| Semi-Supervised | SVM Baseline | - | - | I3D+VGGish | 50.78 | - | - |
| | Sun et al. (2020) | MM 20' | - | | - | 74.70 | 72.7 |
| | Conv-AE (Hasan et al., 2016) | CVPR 16' | - | I3D+VGGish | 30.77 | - | 50.60 |
| | MULDE (Micorek et al., 2024) | CVPR 24' | - | Hiera-L | - | 81.3 | 78.50 |
| Weakly-Supervised | MIL-Rank (Sultani et al., 2018) | CVPR 18' | - | C3D RGB | 73.20 | 86.30 | 75.41 |
| | CA-VAD (Chang et al., 2021) | TMM 21' | - | I3D RGB | 76.90 | 92.25 | 84.62 |
| | RTFM (Tian et al., 2021) | ICCV 21' | - | I3D RGB | 77.81 | 97.21 | 84.30 |
| | CRFD (Wu & Liu, 2021b) | TIP 21' | - | I3D RGB | 75.90 | 97.48 | 84.89 |
| | MSL (Li et al., 2022) | AAAI 22' | - | VideoSwin | 78.59 | 97.32 | 85.62 |
| | S3R (Wu et al., 2022) | ECCV 22' | - | I3D RGB | 80.26 | 97.48 | 85.99 |
| | CMA-LA (Pu & Wu, 2022) | ICCECE 22' | - | I3D+VGGish | 83.54 | - | - |
| | MACIL-SD (Yu et al., 2022) | MM 22' | - | I3D+VGGish | 83.40 | - | - |
| | MGFN (Chen et al., 2023) | AAAI 23' | - | VideoSwin | 80.11 | - | 86.67 |
| | UR-DMU (Zhou et al., 2023) | AAAI 23' | - | I3D RGB | 81.66 | - | **86.97** |
| | Zhang et al. (2023) | CVPR 23' | - | I3D+VGGish | 81.43 | - | 86.22 |
| | CoMo (Cho et al., 2023) | CVPR 23' | - | I3D RGB | 81.30 | **97.60** | 86.10 |
| | HyperVD (Peng et al., 2024) | arXiv | - | I3D+VGGish | 85.67 | - | - |
| | PEL4VAD (Pu et al., 2024) | TIP 24' | ✓ | I3D RGB | 85.59 | 98.14 | 86.76 |
| | VadCLIP (Wu et al., 2024) | AAAI 24' | ✓ | CLIP | 84.51 | - | 88.02 |
| | HLGAtt (Ghadiya et al., 2024) | CVPR 24' | - | I3D+VGGish | **86.34** | - | - |
| | TPWNG (Yang et al., 2024) | CVPR 24' | ✓ | CLIP | 83.68 | - | 87.79 |
| Frame-Supervised- | **Ours** | - | - | I3D RGB | 88.12 | **98.52**(+0.92) | **87.57**(+0.60) |
| | **Ours** | - | - | I3D+VGGish | **88.39**(+2.05) | - | - |

Table 2: FAR comparison with SOTA methods on three datasets.

| Supervision | Methods | Text Annotation | Feature | XD(%) | SH(%) | UCF(%) |
|---|---|---|---|---|---|---|
| Semi-Supervised | Conv-AE (Hasan et al., 2016) | CVPR 16' | - | - | - | - | 27.2 |
| | GODS (Wang & Cherian, 2019) | ICCV 19' | - | BoW+TCN | - | - | 2.10 |
| Weakly-Supervised | MIL-Rank (Sultani et al., 2018) | CVPR 18' | - | C3D RGB | - | 0.15 | 1.90 |
| | GCN (Zhong et al., 2019) | CVPR 19' | - | TSN RGB | - | - | **0.10** |
| | AR-Net (Wan et al., 2020) | ICME 20' | - | I3D RGB | - | 0.10 | - |
| | MIST (Feng et al., 2021b) | CVPR 21' | - | I3D RGB | - | 0.05 | 0.13 |
| | CRFD (Wu & Liu, 2021a) | TIP 21' | - | I3D RGB | - | - | 0.72 |
| | UR-DMU (Zhou et al., 2023) | AAAI 23' | - | I3D RGB | 0.65 | - | - |
| | PEL4VAD (Pu et al., 2024) | TIP 24' | ✓ | I3D RGB | 0.75 | 0.00 | 0.43 |
| Frame-Supervised- | **Ours** | - | - | I3D RGB | 0.50 | **0.00** | 0.40 |
| | **Ours** | - | - | I3D+VGGish | **0.49** | - | - |

A lower FAR indicates more reliable anomaly detection.

encoder to acquire audio features. The $\sigma$ is set as 0.48, 0.5, 0.7 for SF-XD-Violence, SF-UCF-Crime and SF-ShanghaiTech respectively. More implementation details are presented in Appendix B.

### 4.6 COMPARISONS WITH SOTA METHODS

To demonstrate the effectiveness of our method, the proposed method is compared with SOTA Semi-supervised VAD (S-VAD) methods (Sun et al., 2020; Hasan et al., 2016; Micorek et al., 2024) and Weakly-supervised VAD (W-VAD) methods, e.g., (Sultani et al., 2018; Feng et al., 2021b; Park et al., 2023; Pu et al., 2024; Ghadiya et al., 2024; Yang et al., 2022), as depicted in Table 1. Noticed our method delivers superior performance, compared to S-VAD methods and W-VAD methods, and even to the latest methods (Pu et al., 2024; Wu et al.,

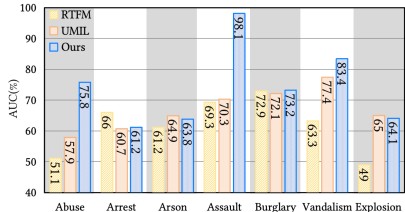

Figure 4: AUC results w.r.t. sub-classes on SF-UCF-Crime

2024; Yang et al., 2024) that trained with additional text annotations of the anomalies. In the challenging and largest SF-XD-Violence, the dataset contains various scenarios, anomaly behaviors, and graphic effects, which bring substantial obstruction to an accurate detection. Even so, our method delivers superior performance, achieving a 2.05% absolute increment in terms of AP compared to the W-VAD methods without text prompt learning and outperforming all recent methods that are augmented with text annotations. The results demonstrate the validity of proposed SAPM module, which realizes accurate anomaly learning with similarity-based abnormal probability and abnormal frame mining algorithm. As a fixed-view campus surveillance dataset, the scenes in SF-ShanghaiTech contain less noise and the behaviors are evident. Our method surpasses all previous methods. With single frame supervision, the model gains reference to precise abnormal behavior,

thereby enabling accurate abnormal pattern modeling. Finally, in SF-UCF-Crime, where anomalies are highly diverse, our approach achieves a promising performance and demonstrates excellent generalization by accurately modeling diverse abnormal behaviors. The strong performance across all datasets can be attributed to our precise frame mining algorithm, similarity-based abnormal probability, and the calibrated TTM module, which helps capture key temporal characteristics to refine detection results. Moreover, we conduct performance comparisons on each type of anomaly with RTFM (Tian et al., 2021) and UMIL (Lv et al., 2023), as depicted in Fig. 4. The results manifest the validity of SF-VAD and our method achieves a superior generalization ability across diverse abnormal behaviors.

To further validate the reliability of our method, we compare its False Alarm Rate (FAR) with state-of-the-art S-VAD methods (Hasan et al., 2016; Wang & Cherian, 2019) and W-VAD methods (Sultani et al., 2018; Feng et al., 2021b; Zhou et al., 2023), as shown in Table 2. Our method achieves the lowest FAR in SF-XD-Violence and SF-ShanghaiTech, demonstrating its superior ability to distinguish normal patterns, leading to more reliable anomaly detection. This improvement is largely attributed to the proposed GNPM module, which effectively decouples normal patterns from abnormal ones guided by Gaussian-prior normal probability, significantly reducing false alarms.

Table 3: Ablation studies of modules on SF-UCF-Crime.

| SAPM | GNPM | TD | BR | AUC(%) | FAR (%) |
|------|------|-----|-----|--------|---------|
| -    | -    | -   | -   | 83.67  | 0.62    |
| ✓    | -    | -   | -   | 85.36  | 0.57    |
| ✓    | ✓    | -   | -   | 86.97  | 0.41    |
| ✓    | ✓    | ✓   | -   | 87.02  | 0.40    |
| ✓    | ✓    | ✓   | ✓   | 87.57  | 0.40    |

Table 4: Ablation studies of normal buffer on SF-UCF-Crime.

| $g$ | 5 | 6 | 7 | 8 | 9 | 10 |
|-----|-----|-----|-----|-----|-----|-----|
| AUC(%) | 85.96 | 86.47 | 87.57 | 86.21 | 85.84 | 85.35 |

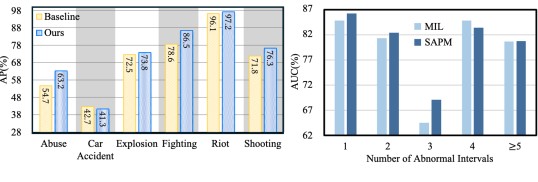

(a) Ablations w.r.t. classes of anomalies (b) Ablations w.r.t. number of anomalies

Figure 5: Ablation studies in SF-XD-Violence

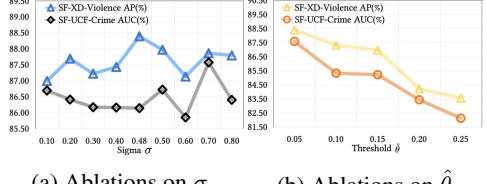

(a) Ablations on $\sigma$     (b) Ablations on $\hat{\theta}$

Figure 6: Ablation studies on parameters in GNPM

### 4.7 ABLATION STUDIES

**Effect of proposed module.** To demonstrate the effectiveness of proposed modules, we conduct ablation experiments on SF-UCF-Crime dataset, as shown in Table 3. The baseline is trained with MIL as depicted in Sec. 3.3. With SAPM, the AUC performance increases 1.69% which proves that proposed SAPM ensures an accurate frame mining and learning strategy guided by similarity-based abnormal probability enabling reliable abnormal pattern modeling. By GNPM, the model decouples normal patterns in abnormal videos. The results not only manifest a 1.61% increment in terms of AUC but also demonstrate 0.16 % decrease of FAR, exhibiting a more reliable detection capability. With temporal decoupling module and boundary refining module, the model achieves a 0.6% improvement in terms of AUC. With the synergy of SAPM and GNPM and calibrated TTM module our method exhibits an 87.57 % AUC. In addition, we evaluate the effect of proposed method on each category of anomaly in SF-XD-Violence as shown in Fig. 5a. Our method surpasses baseline model in almost every category which demonstrates the robust learning of diverse abnormal patterns. Furthermore, we perform an ablation study to compare the detection performance of the baseline model and the model trained with SAPM on videos containing different numbers of anomalies. As Fig. 5b manifests, our method outperforms MIL baseline on almost all videos that contain varied number of anomalies. It verifies that reliable abnormal pattern modeling can enhance the detection of multiple anomaly segments.

**Effect of parameters in GNPM.** We conduct ablation studies on the parameters in GNPM to validate the effectiveness of the normal pattern disentanglement. As shown in Fig. 6a, the model

achieves 86.99% AP in SF-XD-Violence when $\sigma$ equals 0.1 and reaches 88.39% AP with $\sigma$ of 0.48 which proves that the variance of normal distribution contributes to the detection accuracy. The results illustrate that we can boost the detection performance and decouple normal features with proper $\sigma$. In addition, we execute ablation studies on the normal buffer $g$ as depicted in Table 4. The results illustrate that $\theta$ of 7 obtains the best performance of 87.57%, which proves that GNPM can facilitate the normal pattern disentanglement effectively.

**Effect of threshold in SAPM.** To illustrate the effectiveness of the abnormal frame mining module, we conduct ablation studies with varied coherence threshold $\hat{\theta}$, as shown in Fig. 6b. Notably, SAPM achieves the best performance with $\hat{\theta}$ of 0.05 which shows that control of threshold $\hat{\theta}$ can ensure reliable abnormal frame mining.

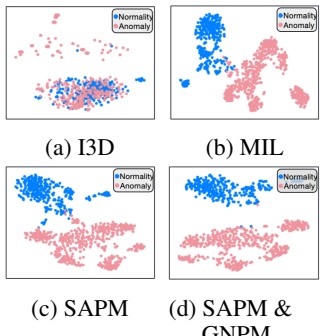

(a) I3D  (b) MIL

(c) SAPM  (d) SAPM & GNPM

Figure 7: Feature distributions under different losses

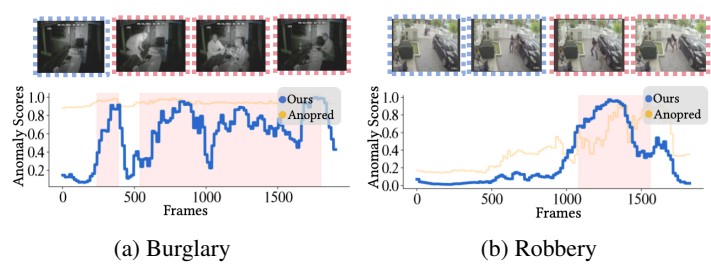

(a) Burglary  (b) Robbery

Figure 8: Qualitative results on SF-XD-Violence. The pink square indicates ground-truth abnormal frames. The Y-axis represents anomaly scores, while the X-axis represents the frames.

### 4.8 QUALITATIVE RESULTS

**Feature Distribution.** To exhibit the capability of proposed method in separating abnormal and normal features for precise anomaly detection, Fig. 7 visualizes the feature distribution of semantic features. Fig. 7a displays the distribution of the original I3D feature which exhibits a clustered and entangled character between abnormal features and normal features. Fig. 7b is the semantic features learned by the MIL which shows some superposition over the feature space. The semantic features guided by SAPM are manifested by Fig. 7c as a more discriminative character is revealed. With the synergy of SAPM and GNPM, the model learns a distinct semantic feature as a clear margin is uncovered in Fig. 7d, the property of which facilitates precise anomaly localization.

**Anomaly Scores.** To substantiate the effect of our method intuitively, the predicted anomaly scores of hard cases are visualized on the challenging SF-UCF-Crime and SF-XD-Violence dataset in Fig. 8, compared to baseline (Liu et al., 2018b). As illustrated in Fig. 8a, our method can detect fine-grained anomalies in long video sequences and accurately distinguish subtle normal patterns between abnormal events. Fig. 8b exemplify the proficiency of our method in capturing trivial abnormal behaviors. In addition, with the enhancement of temporal decoupling module and boundary refining module, our method can disentangle the abnormal and normal features precisely and generate clear abnormal event boundaries. More qualitative results are depicted in Appendix G.

## 5 CONCLUSION

In this paper, we propose single frame supervised VAD (SF-VAD) with three SF-VAD datasets. SF-VAD provides additional dual references to decouple contrary patterns. Correspondingly, we propose Frame-guided Probabilistic Contrary Learning (FPCL). FPCL effectively captures comprehensive abnormal patterns from mined frames, guided by similarity-based abnormal probability for precise learning. It also disentangles normal patterns in abnormal videos by learning from preceding frames, utilizing Gaussian-prior normal probability as guidance. During inference, we introduce temporal decoupling and boundary refinement modules to enhance the discriminative temporal temporal features for refined detection. Extensive experiments demonstrate that our SF-VAD method achieves promising results and delivers an ideal balance between performance and annotation cost.

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

# A  BASELINE

The architecture of the overall framework is depicted in Fig. 2. Concretely, given an untrimmed video with corresponding audio, pertained feature encoder backbones are employed to obtain multi-modal features. Subsequently, the features are passed through the Transformer-based Temporal Modeling (TTM) module and detector to predict frame-level anomaly scores. In training, SAPM is utilized to learn comprehensive abnormal patterns and GNPM is utilized to facilitate fine-grained abnormal representation learning. In inference, we calibrate the attention map in TTM by temporal decoupling and boundary refining module to reveal the discriminative abnormal character of temporal features.

Considering the trade-off of computational overhead and detection performance, the input videos and audio are split into 16-frame non-overlapping clips. Pre-trained frozen encoders are utilized to extract embedding features, formulating clip feature sequences. Video and audio features are then concatenated as multi-modal feature sequence $\mathcal{X} \in \mathbb{R}^{L \times D_m}$ where $L$ equals the number of the clips and $D_m$ is the dimension of the features.

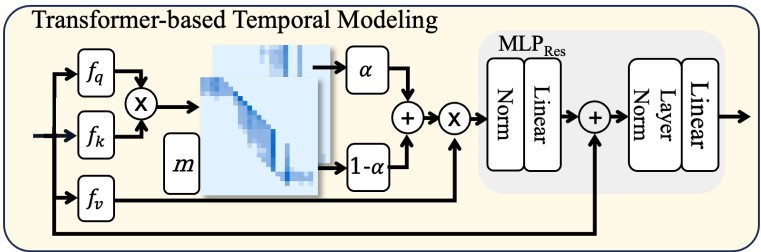

Figure 9: Illustration of Transformer-based temporal modeling module.

Owing to resounding success in natural language processing areas, Transformer (Vaswani et al., 2017) has been verified as a highly effective architecture for capturing global dependencies. And it has been successfully employed in temporal modeling (Yu et al., 2020; Aksan et al., 2021). Therefore, we apply TTM module, following (Pu et al., 2024). As depicted in Fig. 9, TTM captures the temporal relationship of abnormal events, utilizing self-attention mechanism with a multi-scale self-attention based method.

First, the similarity matrix $\mathcal{M} \in \mathbb{R}^{L \times L}$ is computed where dynamic position encoding $\mathcal{E} \in \mathbb{R}^{L \times L}$ is added to incorporate temporal position prior:

$$\mathcal{M} = f_q(\mathcal{X}) \cdot f_k(\mathcal{X})^\top + \mathcal{E}$$
$$\mathcal{E}_{j,k} = \exp\left(-\left|\gamma(j-k)^2 + \beta\right|\right) \tag{15}$$

where $f(\cdot)$ refers to linear layers and $j, k \in [1, L]$ indicate index of clips. $\gamma$ and $\beta$ represent learnable weight and bias. Global attention feature $\mathcal{F} \in \mathbb{R}^{L \times D_h}$ is computed based on the similarity matrix and the linear projection of $\mathcal{X}$. The process can be denoted as follows:

$$\mathcal{F} = \text{softmax}\left(\frac{\mathcal{M}}{\sqrt{D_h}}\right) \cdot f_v(\mathcal{X}), \tag{16}$$

where $D_h$ indicates the hidden dimension. To highlight short-range temporal attention of events and solve long-range noise, the similarity matrix is masked by a sliding window. The process can be denoted as:

$$\widetilde{\mathcal{M}}_{ij} = \begin{cases} \mathcal{M}_{ij}, & j \in \left[\max\left(0, i - \lfloor \frac{w}{2} \rfloor\right), \min\left(i + \lfloor \frac{w}{2} \rfloor, L\right)\right] \\ -\infty, & \text{otherwise} \end{cases} \tag{17}$$

where $w$ refers to the window size and $\widetilde{\mathcal{M}}$ indicates local similarity matrix. Correspondingly, local attention feature $\widetilde{\mathcal{F}} \in \mathbb{R}^{L \times D_h}$ is computed by Eq. 16. Then, global and local features are fused by gate weight $\alpha$. Subsequently, a residual connection is utilized followed by layer normalization to derive temporal feature $\mathcal{F}^t \in \mathbb{R}^{L \times D_m}$, which can be formulated as:

$$\mathcal{F}^t = f_o\left(\text{Norm}\left(\alpha \cdot \mathcal{F} + (1 - \alpha) \cdot \widetilde{\mathcal{F}}\right)\right)$$
$$\mathcal{F}^c = \text{LayerNorm}\left(\mathcal{X} + \mathcal{F}^t\right) \tag{18}$$

where $\mathrm{Norm}(\cdot)$ denotes a composite of power normalization (Yu et al., 2017) and L2 normalization. Eventually, TTM acquires multi-scale temporal feature $\mathcal{F}^c \in \mathbb{R}^{L \times D_m}$ .

Eventually, a MLP-based anomaly detector is employed to predict the frame-level anomaly scores.

## B  IMPLEMENTATION DETAILS

**Feature Extraction.** To extract video and audio features, we follow existing methods (Wu & Liu, 2021b; Pu et al., 2024; Wu et al., 2020). We apply the I3D (Carreira & Zisserman, 2017) video encoder that is pre-trained on Kinetics (Kay et al., 2017) dataset, to acquire video features. I3D processes each video frame and aggregates temporal context over a sequence of frames, enabling it to extract rich, motion-aware features from the video. Video features are extracted from *global_pool* layer from the I3D encoder which is 1024 dimensions. To acquire audio features, we utilize VG-Gish (Hershey et al., 2017) audio encoder that is pre-trained on Youtube (Hershey et al., 2017) dataset. VGGish extracts high-level audio features that capture acoustic patterns, The acquired audio features contain 128 dimensions. For the trade-off of detection performance and computational overhead, each video is split into 16-frame non-overlapping clips, formatting video and audio sequences. Notably, we employ an augmentation strategy to enhance the generalization ability. For SF-UCF-Crime and SF-ShanghaiTech datasets, we apply a ten-crop augmentation strategy which includes crops from the center, four corners, and their mirrored counterparts. For SF-XD-Violence dataset, we employ a five-crop augmentation strategy, which includes crops from the center and four corners.

**Hyperparameter.** The hidden dimension $D_h$ of transformer-based temporal modeling module is set to 128. The initial gate weight $\alpha$ of transformer-based temporal modeling module is set to 0.5. The window size $w$ is set to 5, 9, 9 for SF-ShanghaiTech, SF-UCF-Crime and SF-XD-Violence respectively. The kernel size and stride of the one-dimension convolutional layer $f_t$ are set to 3 and 1 respectively. In SAPM, $\hat{\theta}$ is set to 0.05. In GNPM, the $\sigma$ of the normal probability distribution is set to 0.5, 0.48, 0.7 for SF-ShanghaiTech, SF-UCF-Crime and SF-XD-Violence respectively. Such settings of hyperparameter are obtained by hyperparameter tuning. Interestingly, we find that such a setting is coherent with the statistics of datasets, as depicted in 4.2. As the variance of SF-UCF-Crime is relatively larger than SF-ShanghaiTech, a large $\sigma$ can force the model to focus on a larger portion of normal frames. In the boundary refining module, $\theta'$ and $\theta''$ are set to 4 and 10. In the temporal decoupling module, threshold $\theta$ is set to 0.0015.

**Training Details.** During training, the model parameters are initialized by Xavier initialization. The batch size is set to 64, 128, 128 for SF-ShanghaiTech, SF-UCF-Crime, and SF-XD-Violence respectively. The learning rate is $5 \times 10^{-4}$ initially and controlled by a cosine decay strategy. The parameters are optimized using Adam optimizer. The number of training epochs is set to 50. For the balance between computational overhead and detection performance, the maximum sampling sequence length is set to 200 during the training phase.

## C  DATASET CONSTRUCTION

To adapt single-frame supervision for video anomaly detection (VAD), one of the primary challenges is the absence of appropriate datasets annotated with fine-grained frame-level labels. Most existing VAD research relies on weak supervision, leading to the development of W-VAD datasets. However, the ground truth in W-VAD datasets typically consists of binary, video-level labels, which lack the necessary granularity for single-frame supervision.

To address this, we construct three high-quality, human-annotated Single Frame supervised VAD (SF-VAD) datasets based on well-known W-VAD datasets: ShanghaiTech Campus (Liu et al., 2018a), UCF-Crime (Sultani et al., 2018), and XD-Violence (Wu et al., 2020). These SF-VAD datasets are annotated through a carefully designed crowdsourced annotation process.

We employed 12 human annotators for this task, each selected through a rigorous process. Before starting, annotators had to familiarize themselves with the definitions of various abnormal behaviors, such as violence, accidents, and theft, and then pass a preliminary annotation test to ensure accuracy. Each annotator worked independently, and cross-validation was conducted to ensure the consistency and quality of the annotations. To streamline the annotation process and ensure high accuracy, we

provided annotators with the following guidelines: 1) watch the video until they are absolutely sure that they witness an abnormal frame of any form; 2) annotate the abnormal frame as soon as possible under certain confidence to the existence of anomaly.

Once all individual annotations were complete, a cross-verification process was performed to identify inconsistencies. Discrepancies between annotations were reviewed and corrected, ensuring the final annotations accurately reflect the frames where abnormal events occur.

## D  DATASET DESCRIPTION

In this work, we apply three widely used W-VAD datasets, which contain various abnormal behaviors in diverse scenes.

**ShanghaiTech Campus** (Liu et al., 2018a) comprises 437 videos from 13 fixed-view campus surveillance cameras. The abnormal types are cycling, chasing, cart, fighting, skateboarding, vehicle, running, jumping, wandering, lifting, robbery, climbing over, throwing. The background of the frames is rather steady and contains less noise, which highlights the behaviors within the frames.

**UCF-Crime** (Sultani et al., 2018) comprises 1900 videos collected from a variety of sources including videos from surveillance cameras and social media with a total duration of 128 hours. The dataset covers 13 real-world anomalies of crimes including abuse, arrest, arson, assault, burglary, explosion, fighting, road accident, shooting, shoplifting, stealing, vandalism and robbery. The representations of the anomalies are varied and differentiated which increases the challenge of the detection by requiring a more thorough understanding of the anomaly semantics.

**XD-Violence** (Wu et al., 2020) is the largest and most challenging multi-modal VAD dataset containing 4754 untrimmed videos with a total duration of 217 hours. The dataset contains videos from various sources such as movies, social media, car cameras, surveillance, and games where exist extensive artistic expressions such as changing perspective, view zooming, dynamic lighting, and rapid camera movements. The above characteristics of the datasets draw non-negligible difficulty to anomaly detection models. It covers anomalies of 6 types including abuse, car accidents, explosions, fighting, riots, and shooting.

## E  ANNOTATION TIME ESTIMATION

Data annotation is actually an intricate process that involves both labeling time and hidden time. Labeling time consists of the time that annotators watch the videos, the time that the annotators playback to find a particular frame or make sure the temporal boundary of anomalies, and the time the annotators that double check their annotations. While there is also a huge amount of hidden time, e.g., the time that the supervisors cross validate the annotations and handle the conflicts, the time to debug the annotation platform, or the time of training the annotators to pass the preliminary test. Owing to the complexity of estimating the actual annotation time, in this work, we estimate the annotation time based on the theoretical low bound of estimated annotation time.

**Fully-supervised VAD** utilizes frame-level labels, which requires the annotators to watch the all videos from the beginning to the end at least one time. Thus, the low bound of annotation time equals the entire duration of the datasets.

**Semi-supervised VAD** leverage normal videos only, however, the annotators need to watch the entire video snippets to make sure that the videos do not contain anomalies of any form. Therefore, the low bound of annotation time equals the total duration of the normal videos in the dataset.

**Weakly-supervised VAD** uses video-level binary labels. For videos in the test set, the estimated annotation time is equivalent to the total duration of the test videos. For normal videos in the train set, the estimated annotation time equals the total duration as well, since the annotators need to watch the entire video to make sure it is a normal one. For abnormal videos in the training set, the estimated annotation time is estimated as the sum of time an annotator spends observing an abnormal frame within an abnormal video.

**Single Frame supervised VAD** leverages single frame annotation. Assuming that we elaborately devise an annotation platform, that enables the annotators to label the abnormal frame as soon as they

Table 5: Ablation studies of threshold $\theta'$ on SF-UCF-Crime.

| $\theta'$ | 1 | 2 | 3 | 4 | 5 | 6 |
|---|---|---|---|---|---|---|
| AUC(%) | 86.29 | 86.82 | 87.07 | 87.57 | 87.42 | 87.21 |

Table 6: Ablation studies of threshold $\theta''$ on SF-UCF-Crime.

| $\theta''$ | 7 | 8 | 9 | 10 | 11 | 12 |
|---|---|---|---|---|---|---|
| AUC(%) | 87.26 | 87.48 | 87.57 | 87.46 | 87.27 | 87.07 |

identify one and let annotation proceed, the low bound annotation time is equal to weakly-supervised VAD. Notably, in piratical scenarios, the annotation time of single frame supervised VAD is slightly larger than weakly-supervised VAD, since single frame annotations involve playback from short anomalies and extra cross validation time to handle the conflict of annotations.

## F  HYPERPARAMETER ANALYSIS

**Effect of threshold $\theta'$.** In SF-UCF-Crime, we conduct hyperparameter analysis to illustrate the effect of threshold $\theta'$ which controls the beginning boundary of the anomaly interval in the boundary refining module. As demonstrated in Table 5, when $\theta'$ equals 4, our method achieves the SOTA result which demonstrates that the boundary refining module facilitates generating a clear abnormal event boundary to enable a precise anomaly detection.

**Effect of Threshold $\theta''$.** In SF-UCF-Crime, we further execute the hyperparameter analysis to manifest the effect of threshold $\theta''$ which controls the ending interval of the anomaly interval in the boundary refining module. As illustrated in Table 6, when $\theta''$ is 9, our method can deliver the best result which proves that the proposed boundary refining module can mitigate noisy temporal features, thereby realizing an accurate anomaly detection.

**Effect of window size $w$.** To explore the effect of window size $w$ which controls the local window size in transformer-based temporal modeling module, we conduct experiments with varied $w$ in SF-UCF-Crime. As manifested in Table 7, our method achieves the best detection results with $w$ of 9. The results demonstrate the effect of capturing short-term temporal patterns to solve the long-term noise for precise fine-grained anomaly detection.

## G  QUALITATIVE RESULTS

To illustrate the effectiveness of our method, we visualize the anomaly scores on proposed SF-UCF-Crime and SF-XD-Violence.

In Fig. 10, the blue curves indicate the anomaly scores predicted by our method on SF-UCF-Crime, the yellow curves refer to the anomaly scores predicted by MMIL (Liu et al., 2018a). The pink interval is where the anomaly happens. In SF-UCF-Crime dataset, the videos are shot by fixed-view surveillance cameras. As demonstrated in Fig. 10a, our method manifests the ability to detect long-term anomalies while surpassing the false alarm before the abnormal intervals. Fig. 10b and Fig. 10c further exhibit the capability of our method in localizing anomaly precisely with obscure motion semantics. Fig. 10d displays that our method can recognize normal videos accurately as well.

In Fig. 11 and Fig. 12, the blue curves indicate the anomaly scores predicted by our method on SF-XD-Violence, the yellow curves refer to the anomaly scores predicted by MMIL (Liu et al., 2018a).

Table 7: Ablation studies of window size $w$ on SF-UCF-Crime.

| $w$ | 7 | 8 | 9 | 10 | 11 | 12 |
|---|---|---|---|---|---|---|
| AUC(%) | 87.32 | 87.28 | 87.57 | 87.05 | 86.64 | 86.89 |

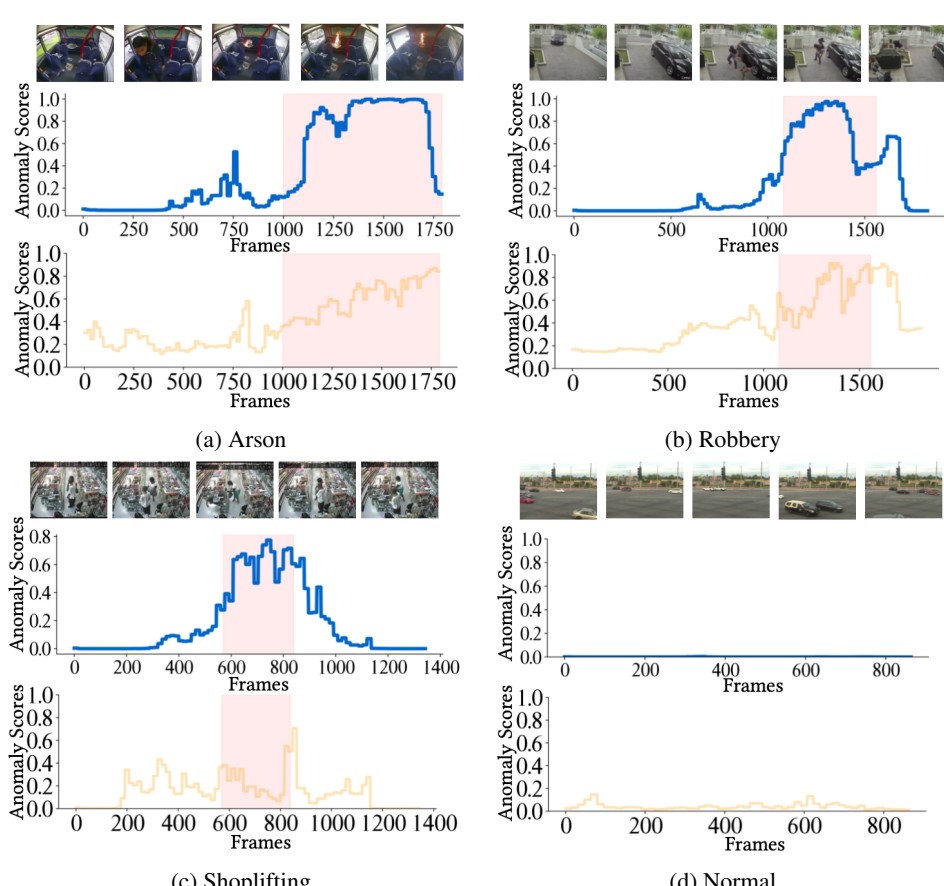

(a) Arson        (b) Robbery

(c) Shoplifting        (d) Normal

Figure 10: Visualization of anomaly scores in the SF-UCF-Crime dataset. The Y-axis displays the anomaly scores, with 1 indicating abnormal and 0 indicating normal, while the X-axis shows the frame numbers of the videos. The pink-shaded regions highlight the frames where anomalies occur. The blue lines represent the predictions from our method, whereas the yellow lines denote the predictions from MMIL (Liu et al., 2018a). The frames above are snapshots from the videos.

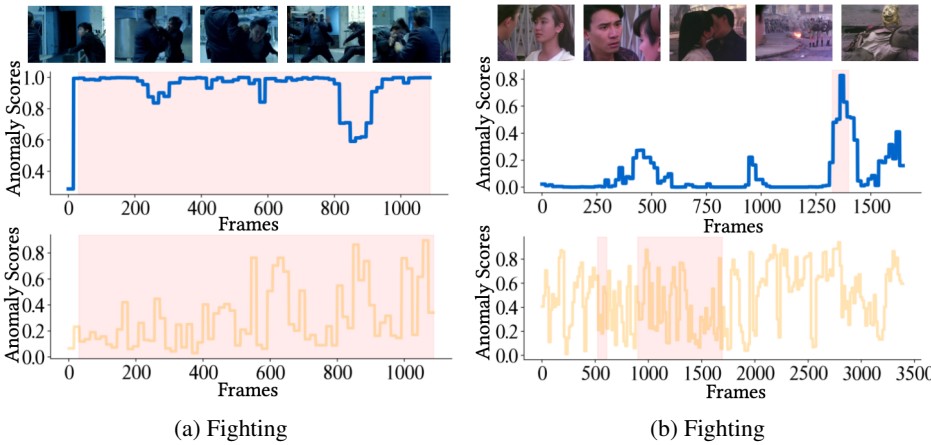

(a) Fighting        (b) Fighting

Figure 11: Anomaly scores of abnormal videos on SF-XD-Violence. The Y-axis represents anomaly scores, while the X-axis represents the frame number of videos.

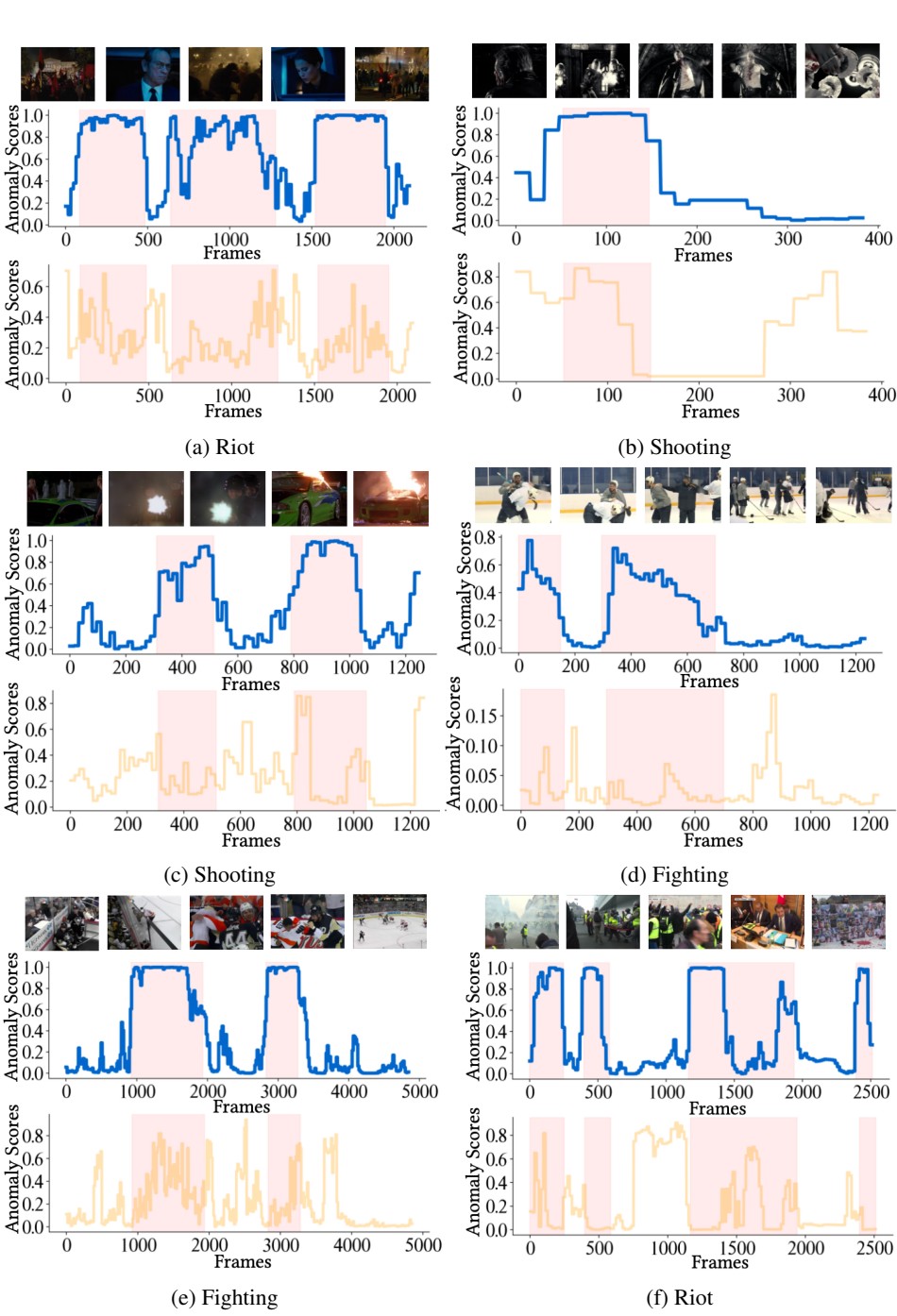

Figure 12: Visualization of anomaly scores for abnormal videos in the SF-XD-Violence dataset. The Y-axis displays the anomaly scores, with 1 indicating abnormal and 0 indicating normal, while the X-axis shows the frame number. The pink-shaded regions highlight the frames where anomalies occur. The blue lines represent the predictions from our method, whereas the yellow lines denote the predictions from MMIL (Liu et al., 2018a). The frames above are snapshots from the videos.

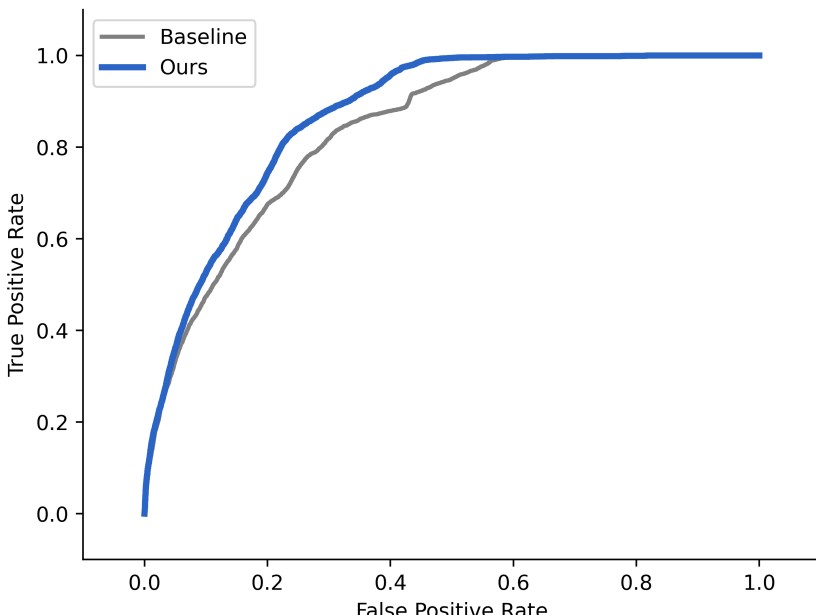

Figure 13: ROC curves of our method and baseline model on SF-XD-Violence. The blue line is the curve of our method, the gray line is the curve of the baseline method trained by MIL.

The pink interval is where the anomaly happens. In SF-XD-Violence dataset, the videos contain a variety of artistic expressions which increases the difficulty of anomaly detection. The behaviors of the videos are also diverse which requires the capability of modeling precise and varied anomaly representations. As demonstrated in Fig. 11a, our method reveals a great ability to capture consistent abnormal intervals. Meanwhile, Fig. 11b manifests that our method can distinguish short-term abnormal behaviors in long-term videos precisely. Fig. 12 further demonstrates the detection performance on some hard cases of SF-XD-Violence including scenes with low lights, drastic scenario change, and crowd disturbance. Fig. 12a shows our approach can detect accurate abnormal behaviors while recognizing short normal intervals within abnormal videos. Fig. 12b manifests that our method can also recognize abnormal behaviors in black-and-white videos. The above results show that with the mutual enhancement of SAPM and GNPM, our method can learn intact fine-grained abnormal representations which facilitate precise abnormal detection and generate clear abnormal boundary. With the mutual enhancement of SAPM and GNPM, our method learns fine-grained abnormal representations. As demonstrated in Fig. 12, our method can not only detect fine-grained abnormal behaviors but distinguish the short intervals as well.

## H  ROC CURVES

Fig. 13 demonstrates the ROC curves of our method and the baseline model on SF-XD-Violence. The blue line is the curve of our method, and the gray line is the curve of our baseline model trained by MIL. The AUC score of our method is 87.57% which surpasses previous SOTA methods. The detection performance is considerably elevated by proposed method. With the advancement of SAPM and GNPM, the proposed method learns fine-grained abnormal representations. As a result, our method can not only detect subtle abnormal behaviors in long-term videos but also recognize normal interval within the abnormal events to mitigate false alarms.

