# OpenReview forum: "Video Anomaly Detection via Single Frame Supervision"
_ICLR.cc/2025/Conference — Submitted to ICLR 2025_

### Official Review · Reviewer_SBUx · 2024-10-28

**Soundness:** 2
**Presentation:** 2
**Contribution:** 3
**Rating:** 5
**Confidence:** 3

**Summary:**

This work proposes single-frame supervision for video anomaly detection where for anomalous data, only one frame per training video is annotated as anomaly. The one frame is annotated with the rule of the first anomalous frame seen by the annotators. In this way, we can mine more anomalous frames and normal frames in a more reliable way. With this newly annotated data, the paper propose a new method consisting of Similarity-based Abnormal Pattern Modeling (SAPM) and Gaussian-prior Normal Pattern Modeling (GNPM). Both components exploit the label in the beginning of anomalous data. Frames before the label are likely to be normal and after the label are likely to be abnormal. SAPM use similarity based method to mine more anomalous frames after the label, i.e., if similar then anomalous frame. GNPM use gaussian weighting, such that the frames far away from the label are more likely to be normal, hence given more weighting to be normal. During inference, filtering techniques, i.e., Temporal decoupling module and Boundary refining module are proposed.

**Strengths:**

The idea of using single frame annotation is novel. It has similar/same annotation time as in weakly supervised setting (video-level annotation).

**Weaknesses:**

1. Despite the novelty, the research direction is not as interesting as fully unsupervised setting [a], where learning using both normal and anomalous data without label at all, which doesn't require any annotation cost at all. Having comparisons including the annotation cost and performance side by side can be done.

2. Whether the annotation will be released is not clear/guaranteed.

3. Line 8-10 of Algorithm 1 seems not explained further in the text.

4. The $l$ and $r$ in Line 8-9 of Algorithm 1 seems not be unrelated with $l$ and $r$ in Eq. (9).

5. See questions.

[a] Zaheer, M.Z., Mahmood, A., Khan, M.H., Segu, M., Yu, F. and Lee, S.I., 2022. Generative cooperative learning for unsupervised video anomaly detection. In Proceedings of the IEEE/CVF conference on computer vision and pattern recognition (pp. 14744-14754).

**Questions:**

1. Why Line 2 and 7 in the algorithm 1 are repeated?

2. The filtering in Eq. (13) seems to be too dependent on the location of anomalous data. For example, what if the anomalous data start near the end of video? Or what if it end near the start of the video?

3. Eq. (7) and Line 279: I think instead of distance to the abnormal annotated frame, it is more towards distance to the first frame. The earlier the annotated frame, the weighting near the annotated frame seems to be larger. Why not doing some kind of inverse gaussian based on the distance towards the annotated frame as described?

---

### Official Review · Reviewer_4UKA · 2024-11-02

**Soundness:** 2
**Presentation:** 3
**Contribution:** 2
**Rating:** 3
**Confidence:** 4

**Summary:**

This paper innovatively proposes the SF-VAD (Single Frame Video Anomaly Detection) problem, which uses only one anomalous frame as a label. The authors claim that this labeling method is efficient because it aligns with how humans discern anomalies, eliminating the need to repeatedly watch videos to determine the time boundaries of anomalies. They have constructed and released three SF-VAD datasets for validation and future research.
GNPM and SAPM are proposed for picking the frames which are similar to abnormal ones as supervision, while modeling normal frames with Gaussian distribution. Temporal decoupling module is also proposed for anomaly scores.

**Strengths:**

1. This study explores inexact supervision for VAD, providing an innovative labeling approach that allows for immediate annotation of an anomaly, offering more detailed labeling information than video-level labels. This establishes a new baseline for VAD and will aid future researchers in constructing larger VAD datasets, which is the paper’s primary contribution.
2. The paper introduces **Similarity-based Abnormal Pattern Modeling**, which learns abnormal patterns, and **Gaussian-prior Normal Pattern Modeling**, which derives normal patterns from the Gaussian prior of the previous frame, adapting to the single-frame labeling.
3. During the inference phase, the paper decouples time and refines anomaly boundaries by filtering out early detections to optimize the final Anomaly Score.

**Weaknesses:**

1. The training of SAPM relies on similarity-based judgments, where frame-wise similarity remains steady. Although the inference phase employs a **Boundary Refining Module** to refine boundaries, certain anomalous samples may not exhibit similar features over time, indicating potential areas for improvement in future research.
2. There are concerns that parameters, such as the similarity threshold (theta), may be adjusted through overfitting on the test set. In fact, Figure 6(b) in the paper shows that increasing the threshold θ\thetaθ leads to a decrease in AP. This raises questions about how to determine optimal hyperparameters across diverse datasets; while the normal buffer ggg is fixed, the duration of anomalous events can vary unpredictably.

**Questions:**

I believe that the Single Frame label represents a weakly supervised and unsupervised labeling method that differs from video-level labels, meeting the needs of real-world labeling scenarios. The paper presents the innovative SF-VAD method, which allows for annotation without the necessity of watching the entire video, significantly reducing labeling time. Additionally, it devises frame-guided probabilistic contrary learning to decouple anomalous and normal patterns through single-frame supervision, providing a new baseline for future VAD research.

I still have some unresolved questions:

1. Could the authors clarify why the **Temporal Decoupling Module** is effective, as it appears to manually set low-variance attention maps to zero?
2. How does the **Boundary Refining Module** function? The paper claims it can clarify event boundaries; if so, I would appreciate additional ablation studies on more datasets.

---

> ### Comment · Reviewer_4UKA · 2024-11-26
> **Discussion**
>
> 1. There are concerns that parameters, such as the similarity threshold (theta), may be adjusted through overfitting on the test set. Please clarify the hyperparameter tuning process and show how you ensured the generalizability of their hyperparameter choices across different datasets.
> 2. Please provide some ablation studies demonstrating the effectiveness of Temporal Decoupling Module.

---

### Official Review · Reviewer_ZW2H · 2024-11-04

**Soundness:** 3
**Presentation:** 3
**Contribution:** 3
**Rating:** 3
**Confidence:** 4

**Summary:**

The paper’s main contribution addresses the labeling issue of the Video Anomaly Detection task by annotating a single anomalous frame in abnormal videos, which is less costly than fully-supervised VAD and more precise than weakly-supervised VAD. The authors annotate three main VAD dataset in this way: UCF-Crime, XD-Violence and ShanghaiTech. To leverage the single-frame supervision, the paper proposes Frame-guided Probabilistic Contrary Learning, consisting of two components. The first is Similarity-based Abnormal Pattern Modeling (SAPM), which computes a similarity between the annotated anomalous frame and following frames until the similarity score is above a set threshold. It is important to notice that, while a single frame is annotated, SAPM effectively results in a fixed set of abnormal intervals for each dataset that do not change across epochs.
The second component is Gaussian-prior Normal Pattern Modeling (GNPM), which models the frames preceding the annotated frame as a Gaussian distribution. GNPM accounts for noisy annotations by leaving a buffer between the annotated frame and the preceding frames.
At inference time, the queries of the attention mechanism are masked according to their variance within a boundary, while the temporal features are masked outside of an abnormal interval obtained from the indexes of the maximum value of the summary of the keys of the local attention map.

**Strengths:**

The paper’s structure is very clear and the experimental section seems complete. The annotations obtained for the dataset can be of value to the field. The proposed approach to modeling normal portions of the video and the attention layer calibration method are both interesting contributions, and seem deserving of further investigations.

**Weaknesses:**

The writing of the paper is unclear is some parts (i.e. ”obscure noise” in line 295 or ”obscure temporal features” in line 316). This is particularly evident in Equation 9, where w is not defined or explained in the main paper, but in section F of the Appendix. The single-frame annotation, while cost-effective, is often not sufficient. The majority of the anomlaous videos in the annotated dataset contain multiple instances on an anomalous event or, in the case of XD-Violence, different anomalous events happen in the same video. This means that the model is trained only on one of them while, for example, the standard MIL framework leads to a more dynamic supervision, albeit often over-relying on the most evident anomalous frames and ignoring more subtle ones or wrongly selecting normal frames.

SAMP constructs the abnormal intervals on which the model is trained to recognize anomalies based on the input video features, leading to fixed abnormal intervals for each video. This seems to be suboptimal for some anomalous actions, such as the ”Shoplifting” class of UCF-Crimes, where the anomalous frames are visually very similar to the normal frames. The paper would benefit from a complete investigation on the class-wise performance of the proposed approach on the UCF-Crime dataset, which contains these types of anomalies. The authors only report in Figure 4 a partial class-wise evaluation on this dataset.

GNPM considers as normal the frames that precede the annotated anomalous frame and includes a buffer to account for noisy annotation. It is not clear what happens if the anomalous event starts very early in the video, as is the case for a large portion of the videos in the datasets used in this paper (as shown also in Figure 3a).

The method seems to rely on seven fine-tuned hyperparameters: ˆ(θ) for SAMP, η, σ and the buffer g for GNPM, w for TD, θ′ and θ′′ for BR. All of them, individually, seem to have a conspicuous impact on the overall performance (see Table 4, 5, 6 and 7, as well as Figure 6). This is an important issue of the proposed method.

The qualitative results presented compare the proposed approach to a method published in 2018. It would be best to compare the qualitative performance with a more recent method.

**Questions:**

In line 083, the authors write: ”As annotators typically mark close to the beginning of abnormal events for efficiency, preceding frames can reasonably be considered as normal.”. Given that the three datasets used in the paper are annotated under the guidance of the authors, has this assumption actually been enforced? This is an important point considering the design of the GNPM and SAPM.

Along with the previous observation, recent works have shown that LLMs have good zero-shot capabilities in VAD, as shown by LAVAD [Zanella et al.(CVPR 2024)]. Did the authors try to obtain the single-frame annotations in such a way? If yes, how is the quality of LLM’s annotation compared to human annotators, considering the cost trade-off?

Table 3 shows that the contribution of the refinement components at inference time allows the model to achieve a higher AUC score on UCF-Crime wrt sota, while GNPM, SAMP and the single-frame annotations score similarly to previous sota. Did the authors evaluate the impact of TD and BR on another publicly available model that uses a transformer block (i.e. URDMU)?

The temporal decoupling module masks attention’s queries outside a boundary. Why the queries?
Similarly, it is not clear why it is necessary to use the attention’s keys to mask the features at inference time. In line 318 the authors say that the goal is to ”generate clear event boundaries”, but why the keys instead of the queries?

---

### Official Review · Reviewer_SYDC · 2024-11-04

**Soundness:** 2
**Presentation:** 2
**Contribution:** 3
**Rating:** 5
**Confidence:** 4

**Summary:**

The paper presents a novel approach to Video Anomaly Detection (VAD), specifically addressing the challenges in fully-supervised and weakly-supervised VAD.

**Strengths:**

The idea is novel and the proposed method is of great value.

**Weaknesses:**

- The description of FPCL, particularly how the abnormal and normal probabilities are defined and utilized, could be expanded.
- The authors claim superiority over SOTA methods, but it would be useful to see comparisons with a broader range of baseline methods.
- A more detailed analysis of the temporal decoupling and boundary-refining modules would strengthen the discussion of the inference stage.
- The paper could discuss potential limitations, such as scenarios where single-frame annotations might be insufficient or introduce ambiguity, and how this may impact performance.

**Questions:**

Please see the Weaknesses.

---

### Official Review · Reviewer_hnoM · 2024-11-05

**Soundness:** 3
**Presentation:** 3
**Contribution:** 2
**Rating:** 5
**Confidence:** 4

**Summary:**

This paper enhances video anomaly detection by using single timestamp labels that indicate the start of an anomaly, providing minimal yet effective supervision. Leveraging this prior information, the authors propose Gaussian-prior Normal Pattern Modeling (GNPM) to capture normal patterns in the frames preceding the timestamp within anomalous videos. Additionally, they introduce Similarity-based Abnormal Pattern Modeling (SAPM) to model abnormal patterns effectively based on the single frame annotation. Together, these methods improve the model’s ability to distinguish between normal and abnormal sequences, achieving robust anomaly detection with minimal annotation effort.

**Strengths:**

1. The experimental results show that the proposed method outperforms existing weakly-supervised approaches, demonstrating its effectiveness in video anomaly detection even with minimal supervision.

2. The method is intuitive and easy to understand, making the concepts and implementation accessible. This simplicity, coupled with effective results, underscores its practical applicability.

3. The use of a single timestamp to mark the start of an anomaly is a novel approach that significantly reduces annotation workload, enabling robust anomaly detection with minimal supervision. This innovation makes the method both efficient and practical for real-world applications.

**Weaknesses:**

1. The ablation study shows that AUC scores are sensitive to the parameter settings in both GNPM and SAPM across different datasets, which could indicate a limitation in the generalization ability of the proposed method. This sensitivity suggests that the model may require careful parameter tuning for optimal performance on new datasets.

2. It would strengthen the paper to include comparisons with fully-supervised methods on fully annotated datasets. Since the proposed method relies only on the start frame as an annotation, such a comparison would better illustrate its effectiveness and efficiency. Limiting comparisons to only weakly-supervised and semi-supervised methods leaves an incomplete assessment of the model’s overall performance.

3. There are minor typos in the paper that could be addressed for clarity. For example, in Figure 6, the caption misses “SAPM,” and on line 537, the word “temporal” is repeated.

**Questions:**

See the weaknesses.

---

### Meta-Review · Area_Chair_6T8J · 2024-12-19

**Metareview:**

This paper was reviewed by five experts in the field. The final ratings are 5,5,3,5,3. While reviewers generally agree that the single-frame supervision for video anomaly detection is interesting, they also raised several concerns, such as sensitivity of the algorithm on hyperparameters, unclear explanations in part of the paper, etc. The authors did not provide a rebuttal, so there is no ground to overrule reviewers' recommendations. The decision is to reject.

**Additional Comments On Reviewer Discussion:**

The authors did not engage in the rebuttal.

---

### Decision · Program_Chairs · 2025-01-22

Reject